# Structure guided studies of the interaction between PTP1B and JAK

Rhiannon Morris [1,2], Narelle Keating[1,2], Cyrus Tan[1,2], Hao Chen[1,2], Artem Laktyushin[1], Tamanna Saiyed[1], Nicholas P. D. Liau[1,2], Nicos A. Nicola[1,2], Tony Tiganis [3,4,5], Nadia J. Kershaw [1,2] & Jeffrey J. Babon [1,2✉]

Protein Tyrosine Phosphatase 1B (PTP1B) is the prototypical protein tyrosine phosphatase and plays an essential role in the regulation of several kinase-driven signalling pathways. PTP1B displays a preference for bisphosphorylated substrates. Here we identify PTP1B as an inhibitor of IL-6 and show that, in vitro, it can dephosphorylate all four members of the JAK family. In order to gain a detailed understanding of the molecular mechanism of JAK dephosphorylation, we undertook a structural and biochemical analysis of the dephosphorylation reaction. We identified a product-trapping PTP1B mutant that allowed visualisation of the tyrosine and phosphate products of the reaction and a substrate-trapping mutant with a vastly decreased off-rate compared to those previously described. The latter mutant was used to determine the structure of bisphosphorylated JAK peptides bound to the enzyme active site. These structures revealed that the downstream phosphotyrosine preferentially engaged the active site, in contrast to the analogous region of IRK. Biochemical analysis confirmed this preference. In this binding mode, the previously identified second aryl binding site remains unoccupied and the non-substrate phosphotyrosine engages Arg47. Mutation of this arginine disrupts the preference for the downstream phosphotyrosine. This study reveals a previously unappreciated plasticity in how PTP1B interacts with different substrates.

[1] Walter and Eliza Hall Institute of Medical Research, 1G Royal Parade, Parkville 3052 VIC, Australia. [2] Department of Medical Biology, The University of Melbourne, Royal Parade, Parkville 3052 VIC, Australia. [3] Monash Biomedicine Discovery Institute, Monash University, Clayton, VIC 3800, Australia. [4] Department of Biochemistry and Molecular Biology, Monash University, Clayton, VIC 3800, Australia. [5] Cancer Immunology Program, Peter MacCallum Cancer Centre, Melbourne, VIC 3000, Australia. ✉email: babon@wehi.edu.au

Protein tyrosine phosphatase 1B (PTP1B) is a key regulator of several kinases[1–7]. Its role in the regulation of insulin and leptin signalling has been well studied[1,8], and it is implicated in regulating several pathways dependent on JAK-STAT signalling[7,9]. More recently, there has been an increased interest in the role of PTP1B as a checkpoint in tumor immunity. Deletion or inhibition of PTP1B in T cells and dendritic cells enhances anti-tumor immunity via upregulation of JAK-STAT signalling, leading to T cell and dendritic cell activation and expansion[10–12]. This increase in signalling upon inhibition of PTP1B activity is thought to be due to sustained phosphorylation of signalling components, including JAKs, which PTP1B has previously been shown to negatively regulate[1,7]. Although the mechanism of substrate recognition by the PTP1B phosphatase domain has been described in detail for IRK[7], the association of PTP1B with JAK has not been as well characterized. The JAK and IRK kinase domains have a high level of sequence homology, share a similar mechanism of auto-activation, and both contain tandem phosphotyrosine residues within their activation loop. The structure of the PTP1B phosphatase domain in complex with a bisphosphopeptide corresponding to the IRK activation loop revealed that the first (N-terminal) phosphotyrosine of the activation loop occupies the catalytic pocket, while the adjacent (second) phosphotyrosine crosses a gateway region and interacts with a positively charged patch on the surface of PTP1B referred to as the second aryl binding site[7,13]. This interaction is suggested to provide specificity for tandemly phosphorylated substrates (X-pY-pY-X) such as the IRK and JAK activation loops. Phosphatases have since been categorised based on whether they contain this second aryl binding site, whether it is accessible, and therefore whether they prefer singly or tandemly phosphorylated substrates[13]. Based on this classification, PTP1B is the prototypical member of the first category: the gateway region is open and the second aryl binding site has the capacity to bind a second phosphotyrosine[13].

Here we show that all four JAK kinase domains are dephosphorylated by PTP1B and structurally characterise this interaction using X-ray crystallography. We characterise several catalytically-impaired mutants of PTP1B and discovered that different mutants allow visualisation of bound substrate or the bound products of the dephosphorylation reaction. Unlike for the IRK activation loop, the second phosphotyrosine of JAK2 and TYK2 activation loops occupied the catalytic pocket and the second aryl binding site was empty. Our biochemical analyses confirmed that PTP1B has a preference for dephosphorylation of the second phosphotyrosine in the JAK (but not IRK) activation loop. The first phosphotyrosine was located in a shallow pocket of PTP1B centred around Arg 47 and mutation of this residue alters phosphotyrosine preference. Taken together our findings describe the molecular basis of the PTP1B/JAK interaction and reveal an unappreciated plasticity in the interaction between PTP1B and different substrates.

## Results

**PTP1B negatively regulates IL-6 signalling in M1 cells and dephosphorylates the JAK activation loop.** The M1 murine myeloid leukemic cell line responds in a dose-dependent manner to IL-6 stimulation[14]. Culturing M1 cells in IL-6 results in differentiation from monocytic-like progenitor cells into terminally differentiated macrophages that undergo cell death. Higher concentrations of IL-6 were required to induce differentiation of M1 cells overexpressing PTP1B, with an $EC_{50}$ of approximately 10 ng/mL when compared to 2 ng/mL for WT M1 parental cells (Fig. 1a). This was in contrast to five other phosphatases that were similarly overexpressed. Using an Incucyte growth assay, the growth of wild-type M1 cells was compared to that of M1 cells overexpressing HALO-PTP1B both with and without IL-6 (Fig. 1b). Cells overexpressing PTP1B displayed enhanced growth kinetics in the presence of IL-6, compared to wild-type. As expected, there was no difference in the growth rate of any of these cell lines in the absence of IL-6. Together, these results indicate that PTP1B negatively regulates IL-6 JAK-STAT signalling in M1 cells. To confirm that the activation loop of the JAKs are indeed a target of PTP1B, dephosphorylation assays with activated recombinant JAK kinase domains were performed in vitro. All four members of the JAK family were dephosphorylated in a PTP1B-dependent manner although JAK1 was dephosphorylated to a lesser extent than others (Fig. 1c). Likewise, in 293 T cells engineered to overexpress JAK1, overexpression of HALO-PTP1B induced by doxycycline decreased levels of phospho-JAK1 as seen by western blot (Fig. 1d) indicating PTP1B can dephosphorylate the native JAK activation loop in vitro.

**Biochemical characterisation of catalytically impaired PTP1B mutants.** The catalytic cycle of PTP1B has been well described with the aid of several substrate-trapping mutants[9,15–21] and the contribution of protein dynamics to its catalytic rate is also well established[22,23]. Dephosphorylation of substrate is initiated by nucleophilic attack by the active-site cysteine (Cys215) to form a phosphoenzyme intermediate followed by hydrolysis of the cysteinyl-phosphate to release free phosphate. Asp181 acts as both a proton donor, enabling protonation of the tyrosyl leaving group, and as a proton acceptor, receiving a proton from the water molecule that hydrolyzes the phosphoenzyme intermediate[24]. Mutation of Cys215 to alanine completely abrogates enzymatic activity[20] and has allowed the structures of several PTP1B/substrate complexes to be solved[7]. Mutation of Asp181 to alanine reduces $k_{cat}$ by five orders-of-magnitude[20] and mutation of Gln262 (believed to position the catalytic water for nucleophilic attack) results in a 100-fold decrease in $k_{cat}$[20]. The Q262A mutation has previously been used to determine the structure of the cysteinyl-phosphate intermediate[25] and the double mutant D181A/Q262A has been shown to have superior substrate-trapping properties compared to D181A alone[21]. D181A/Q262A however has not been structurally characterised to date. We focused on D181A/Q262A as our preferred substrate trapping mutant but also generated a C215A/D181A/Q262A triple mutant in case the residual catalytic activity of D181A/Q262A did not allow capture of PTP1B in a substrate-bound state. Biochemical characterisation of these mutants highlighted some interesting differences. As the key characteristic of an effective substrate-trapping mutant is a low dissociation rate constant (i.e a slow off-rate) we used surface plasmon resonance (SPR) to compare off-rates. In these experiments a phosphopeptide corresponding to the JAK3 activation loop was immobilized on an SPR chip and PTP1B protein flowed over it (Fig. 2a, b). Surprisingly, the C215A/D181A/Q262A triple mutant had an off-rate that was more than 1000x slower than either the WT or other mutant proteins. ($k_{off}$: C215A/D181A/Q262A $3.4 \pm 0.4 \times 10^{-5} \, s^{-1}$; D181A/Q262A $5.4 \pm 1.3 \times 10^{-2} \, s^{-1}$; C215A $1.0 \pm 0.1 \times 10^{-2} \, s^{-1}$; WT $2.5 \pm 0.2 \times 10^{-2} \, s^{-1}$; $n = 3$, $\pm$SD). This corresponds to an extraordinarily stable complex with a half-life of >5 h compared to ~1 min for D181A/Q262A or C215A alone.

In order to characterise the association of these PTP1B mutants with substrates further, thermal shift analysis using a JAK1 phosphopeptide containing a non-hydrolyzable diF-phosphotyrosine analogue (JAK(diF)) (Supplementary Fig. S1) was used (Fig. 2c). As expected, titration of JAK(diF) into WT and C215A PTP1B led to a progressive increase in Tm. However, titration of

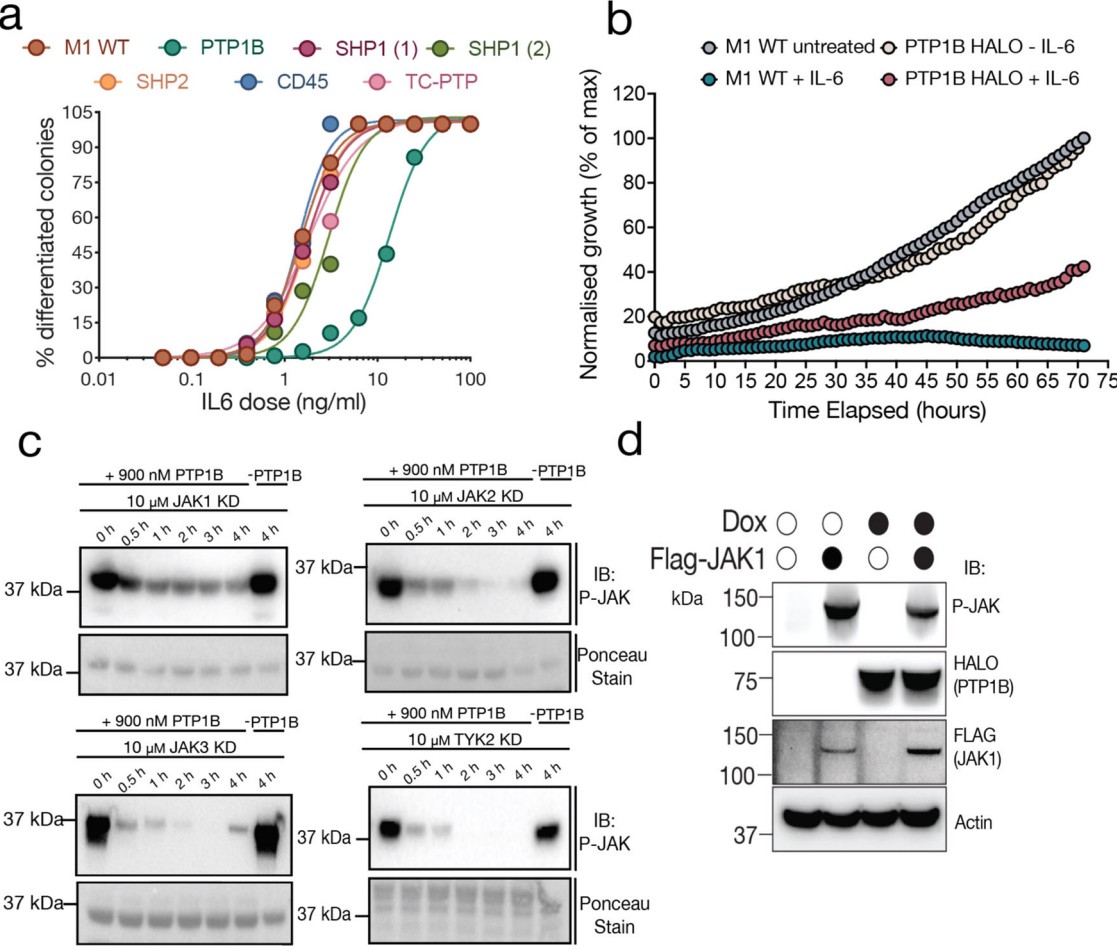

**Fig. 1 PTP1B dephosphorylates the JAK activation loops. a** IL-6 dose response curves for phosphatases overexpressed in the M1 cell line. The effective concentration at which half the maximum response is observed (EC50) for the WT M1 cell line is approximately 2 ng/mL. Overexpression of PTP1B in the M1 cells caused a decrease in response to IL-6, with an EC50 of approximately 10 ng/mL. **b** Growth kinetics of WT M1 cells and PTP1B overexpressing M1 cells in the presence and absence of IL-6 as measured by incucyte. Data displayed are representative of 5 independent experiments. **c** Dephosphorylation assay of the four recombinant JAK kinase domains; JAK1, JAK2, JAK3 and TYK2 by PTP1B (-PTP1B denotes where no PTP1B was added to the reaction). (Top) Immunoblot (IB) probing for phosphorylated-JAK (P-JAK1 antibody) activation loop. (Bottom) Ponceau stained membrane to confirm equal input of proteins on gel. Data are representative of 3 independent experiments. **d** Overexpression of FLAG-JAK and HALO-PTP1B in HEK293T cells. Overexpression of PTP1B leads to a decrease in p-JAK1 when compared to cells not expressing PTP1B. Data displayed are representative of 3 independent experiments.

JAK(diF) into the D181A/Q262A/C215A triple mutant leads to the immediate formation of a highly thermostable species that becomes more populated as the titration progresses. This observation is consistent with the formation of an extremely stable, "covalent-like", complex which does not interconvert with the apo-form during the course of the measurement. Interestingly, the D181A/Q262A mutant exists in two states in the apo-form. The majority of the apo-protein has a Tm of 70 °C (similar to the substrate bound form of the triple mutant), whilst the minor species has a Tm of 55 °C. Titration of substrate leads to a progressive increase in the Tm of the minor species until it all exists in the Tm = 70 °C state.

**PTP1B^D181A/Q262A allows visualisation of both products of the phosphatase reaction in the enzyme active site**. To better understand the interesting thermal shift profile of the D181A/Q262A double mutant, we solved its crystal structure in both apo- and substrate-bound forms (Fig. 3). The apo structure was similar to the apo structure of wild-type PTP1B (PDB: 2CM2[26]) with the one notable exception that the WPD loop (which closes over the top of a substrate tyrosine) was already in its closed position.

Although the WPD loop is usually closed when a substrate is bound, it is noteworthy that the previously determined PTP1B Q262A structure (PDB 1A5Y[25]) also shows the loop in the closed state. It is possible that a preference for being in the closed position, even in apo form, explains the hyper-stable apo-species seen in the thermal shift profile of the D181A/Q262A double mutant (Fig. 2).

To determine the substrate-bound form of the D181A/Q262A double mutant we used the bisphosphorylated activation loop sequence of all four members of the JAK family as substrate peptides (Fig. 3d). Incorporation of the D181A/Q262A mutations leads to a five order-of-magnitude reduction in catalytic activity compared to WT PTP1B[20]. Nevertheless, in multiple co-substrate structures of this mutant, we observed that both steps of the reaction had occurred; dephosphorylation of the pTyr, presumably via the usual cysteinyl phosphate intermediate, as well as hydrolysis of that cysteinyl-phosphate. Surprisingly, the products of the phosphatase reaction (both the phosphate and the tyrosyl-peptide) were clearly visible in the active site in all four structures. The phosphate was not present as a cysteinyl-phosphate intermediate nor as the pTyr substrate but rather as free

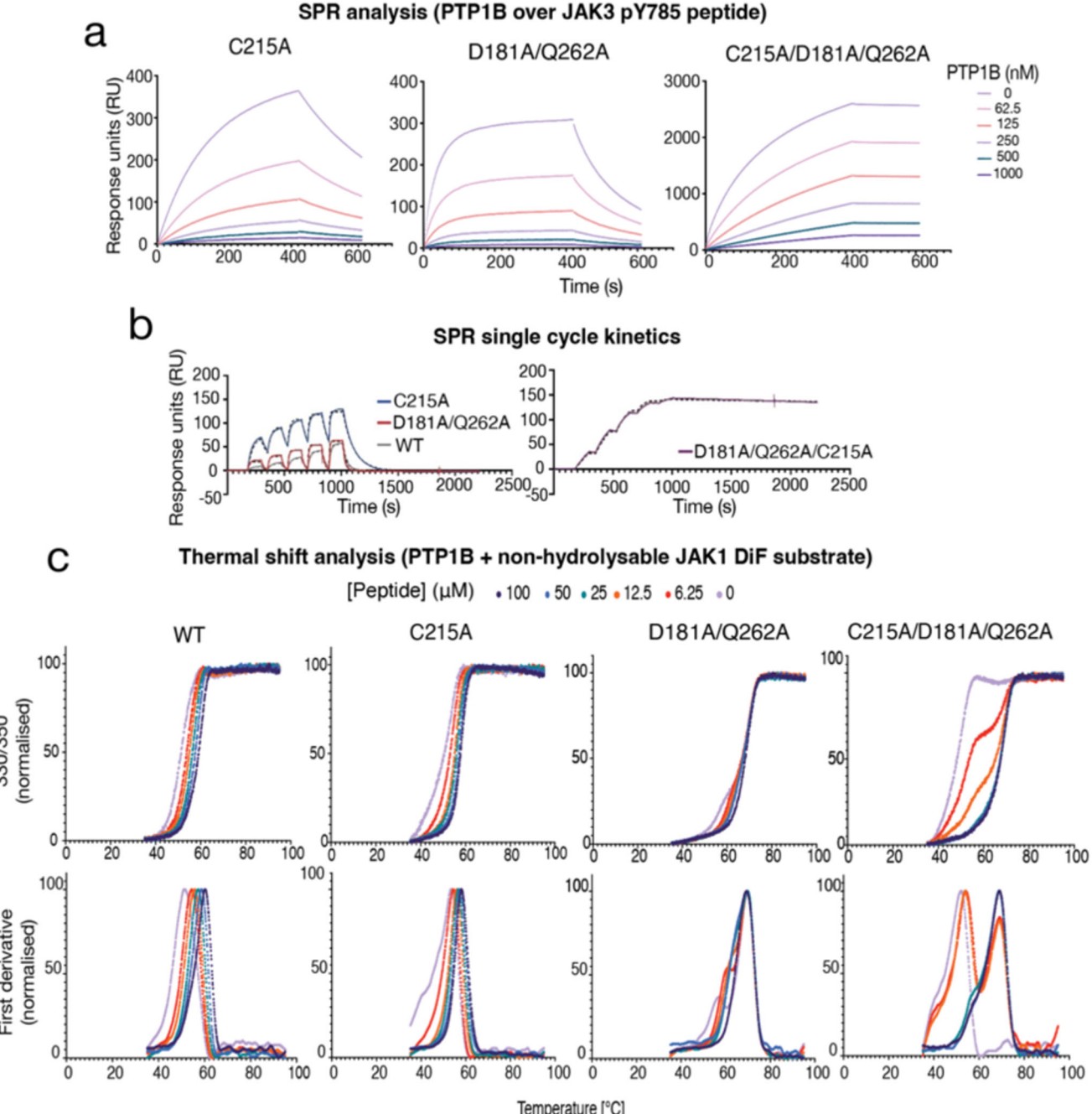

**Fig. 2 Biochemical analysis of catalytically impaired PTP1B mutants. a** SPR analysis of PTP1B mutants binding to immobilised JAK3 pY785 activation loop pTyr peptide. The D181A/Q262A/C215A mutant displays a significantly slower off-rate. Data are representative of 3–4 independent experiments. **b** Quantification of the off-rate of the PTP1B/JAK3 pY785 peptide interaction using single-cycle kinetics. The raw data is shown as solid lines with the fitted curves shown as dashed lines. The off-rate of the triple mutant is 1000x slower than the C215A or D181A/Q262A mutants or WT. **c** Analysis of the PTP1B/ JAK3 pY785 interaction by thermal shift assay. A non-hydrolyzable form of the peptide was used that contains difluorophosphotyrosine (JAK(diF)). The thermal stability of the WT protein increases gradually with the addition of peptide as expected. In contrast, the D181A/Q262A/C215A mutant data indicates a free- and bound-form that do not interconvert during data acquisition. The raw data is shown in the upper panels and the first derivative of that data in the lower panels. Data are representative of 2 or 3 independent experiments.

phosphate. This suggests that dephosphorylation had occurred within the crystal, as well as subsequent hydrolysis of the phosphoenzyme intermediate, but that neither the phosphate nor the tyrosyl-peptide product had completely diffused out of the active site. Accommodating both of these species within the active site necessitates a reorientation of both the tyrosine (back and across, towards solvent) and the phosphate (towards Q266, where it has displaced the water molecule W1 that Q266 coordinates[25])

compared to previous structures[27]. In the wild-type enzyme, the dephosphorylation and cysteinyl-phosphate hydrolysis steps require the loss and gain of a proton, respectively, both mediated by the sidechain of D181, which is absent in our mutant, and also the action of a water molecule as an attacking nucleophile (coordinated by Q262, which is also absent). However, it is unclear which residue takes the place of D181 to act as the general acid/base. The occupancy of the phosphate was refined as ~60%

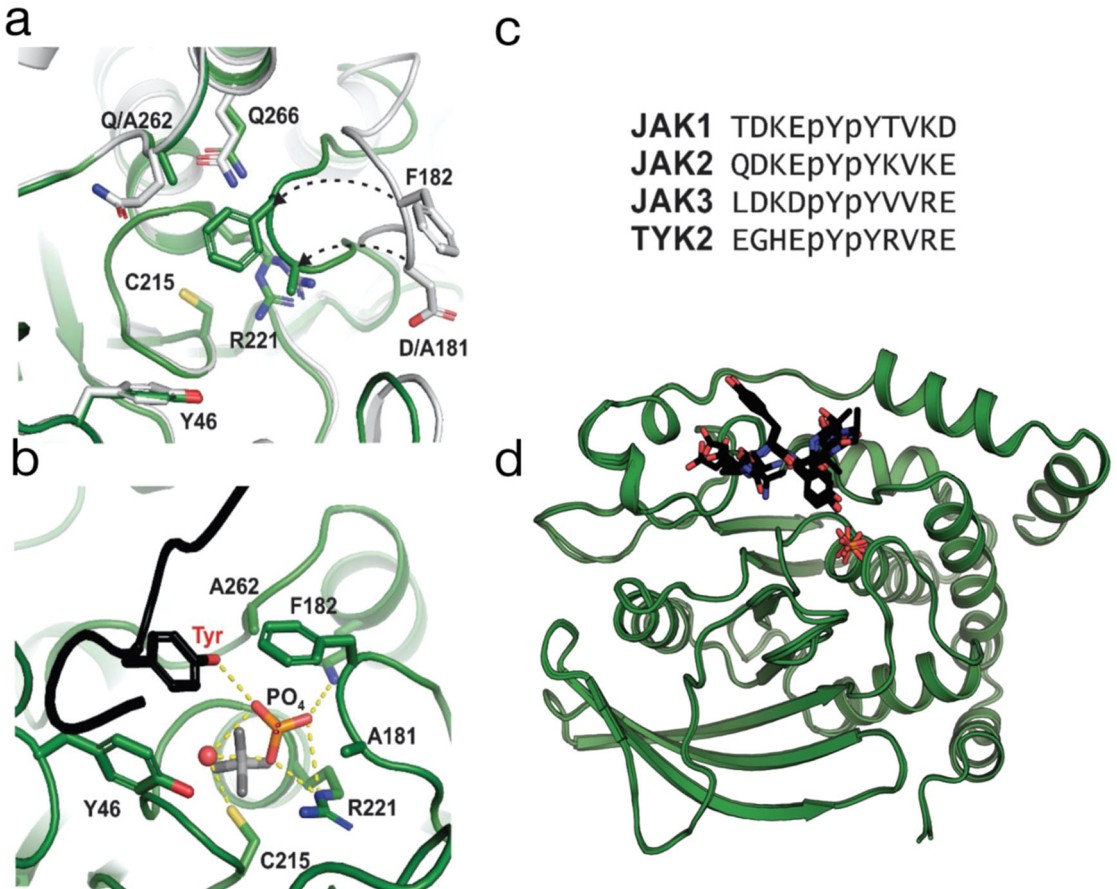

**Fig. 3 PTP1B D181A/Q262A is a product trapping mutant. a** The structure of PTP1B 560 D181A/Q262A (green) shows that the WPD loop is closed even in the apo form, compared to the wild-type protein (white, PDB 2HNQ [28]). **b** Interactions between the PTP1B phosphatase domain, JAK2 activation loop peptide and free phosphate. The free phosphate is coordinated by the side chains of Cys 215, Arg 221, Gln 266 and the substrate tyrosine. The phosphate (orange/red) has shifted in comparison to the tungstate product complex (grey) (PDB 2HNQ [28]) and has displaced a water coordinated by Q266 in the wild-type structure. **c** Sequences of each JAK activation loop peptide used in this study. **d** Overlay of all four structures of PTP1B D181A/Q262A bound to the JAK1, JAK2, JAK3 or TYK2 peptide shows that the product peptide and phosphate remains trapped in the catalytic pocket in every case.

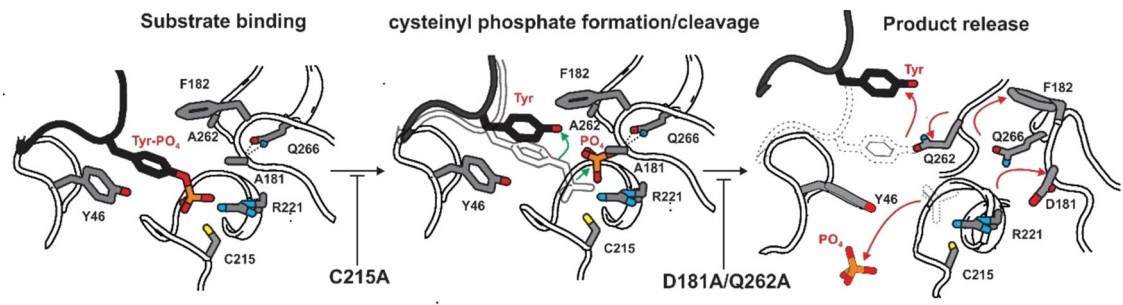

**Fig. 4 Effect on PTP1B mechanism by D181, Q262 and C215 mutations.** When substrate binds to wild-type PTP1B, C215 acts as an attacking nucleophile to form a cysteinyl-phosphate intermediate. Mutation of D181 (the general acid/base) and Q262 (which positions an attacking water molecule for hydrolysis of the cysteinyl-phosphate) slows the reaction rate by several orders of magnitude but do not entirely block cysteinyl-phosphate formation or its subsequent hydrolysis. These mutations also trap the phosphate and tyrosine products of the reaction in the active site, potentially by favouring the WPD loop in its closed conformation.

suggesting that a proportion of it had left the active site but that the majority still remained. Our structure suggests that incorporation of the D181A/Q262A mutations have corrupted the usual catalytic cycle in which substrate is released prior to hydrolysis of the cysteine phosphate (Fig. 4). The substrate tyrosine hydroxyl group in the product-trapped structures occupies the position of the Q262 sidechain in structures of the active enzyme, for example the complex with transition-state

analogue vanadate (PDB 3I80[27]). This sidechain undergoes a change in rotamer between the apo-form and the transition state and this motion may cause it to compete with the substrate tyrosine during catalysis, thereby releasing product.

In some of the JAK activation loop structures with the D181A/Q262A double mutant the register of the peptide in the active site is possibly scrambled, with populations of both the first and second tyrosines being present in the active site (see

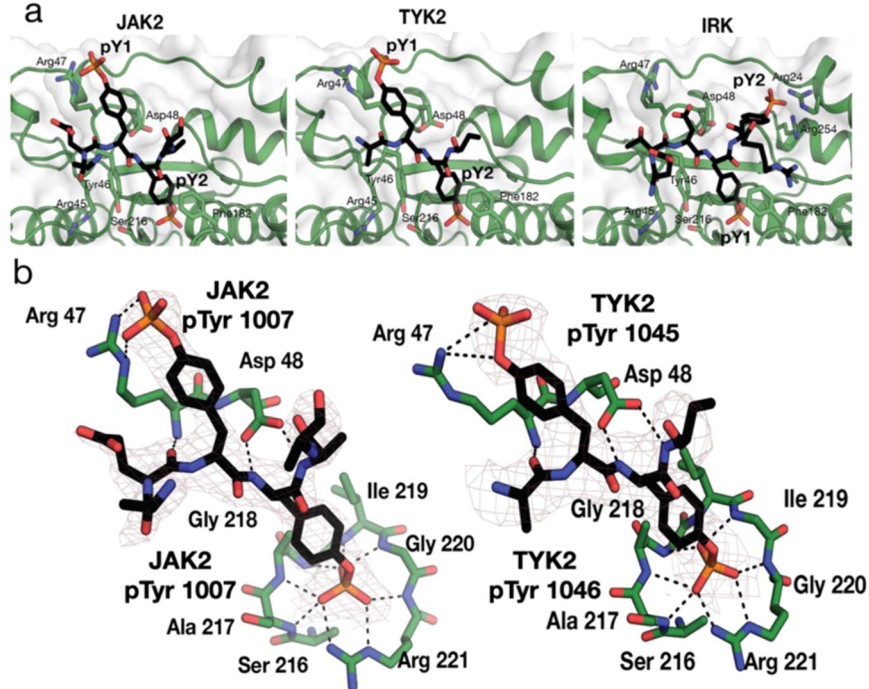

**Fig. 5 The crystal structure of PTP1B in complex with bisphosphorylated JAK peptides indicates a preference for the second phosphotyrosine.**
**a** Structure of PTP1B C215A/D181A/Q262A mutant bound to JAK2 and TYK2 activation loop peptides compared to IRK. Arg 47 of PTP1B interacts with the first phosphotyrosine of both JAK2 and TYK2 (phosphotyrosine 1007 of JAK2 and phosphotyrosine 1045 of TYK2), while the second phosphotyrosine (phosphotyrosine 1008 of JAK2 and phosphotyrosine 1046 of TYK2) sits in the catalytic pocket. **b** Detailed view of the interactions between the JAK peptides and PTP1B with 2Fo-Fc composite omit map with simulated annealing, contoured to 1 σ. Maps are shown only for the JAK2 and TYK2 activation loop peptide residues.

Supplementary Fig. S2). To clarify which pTyr was the preferred substrate, we therefore performed co-crystallisation experiments using the C215A/D181A/Q262A triple mutant which is catalytically dead.

**PTP1B binds the second phosphotyrosine of the JAK activation loop in the catalytic site.** To understand the association of the triple substrate trapping mutant of PTP1B with the JAK activation loops, the co-crystal structures of the PTP1B C215A/D181A/Q262A triple mutant bound to the JAK2 and TYK2 activation loop peptides were solved to 2.01 and 2.79 Å resolution respectively (Supplementary Table 1). Two copies of the PTP1B:JAK2 peptide complex were present in the asymmetric unit. Unexpectedly, in both the JAK2 and TYK2 structures, the second phosphotyrosine occupied the catalytic pocket[7,20,28] (phosphotyrosine 1008 for JAK2 and phosphotyrosine 1046 for TYK2), rather than the first as previously seen for IRK (Fig. 5). In contrast to IRK, we found that the first phosphotyrosine (phosphotyrosine 1007 of JAK2 and phosphotyrosine 1045 of TYK2) in both of these structures was located in a shallow, basic surface near Arg 47. The guanidinium group of Arg 47 forms a salt bridge with the phosphate while the backbones of both amino-acids form hydrogen bonds with one another. In one copy of the PTP1B:JAK2 complex, the density for the guanidinium group of Arg 47 is weak, which suggests some flexibility of this residue. In all models, the sidechain of Asp 48 formed a hydrogen-bond with the backbone amide of the phosphotyrosine in the catalytic site as well and the +1 residue of the peptide (Supplementary Figs. S3, S4). These data differ from the previous observation that PTP1B has a preference for the first phosphotyrosine in the activation loop of IRK[7] (Fig. 5a) and indicates that PTP1B may interact with different substrates via different modes.

**The second pTyr of JAK2 is dephosphorylated more rapidly in the bisphosphorylated substrate.** Although structural data suggested a preference for dephosphorylation of the second pTyr, we sought to confirm this biochemically. A Michaelis-Menten analysis of dephosphorylation of JAK2 and IRK activation loop peptides indicated an enhanced dephosphorylation rate for both peptides when in the bisphosphorylated state (Fig. 6a), consistent with previous data on IRK[7]. For the monophosphorylated peptides, a decrease in $K_M$ and increase in Vmax when the second tyrosine was phosphorylated was observed for JAK but not IRK, consistent with the structural analysis. Using [1]H NMR we then measured the dephosphorylation of individual YpY, pYY and YY peptides over time (Supplementary Fig. S5), and similarly, observed an overall preference for bisphosphorylated peptides, however JAK2 YpY was dephosphorylated at a similar rate as the pYpY peptides, highlighting again the preference for this residue by PTP1B (Fig. 6).

These experiments, however, were not able to explicitly show that the second phosphotyrosine is more rapidly dephosphorylated when part of the bisphosphorylated peptide species. To demonstrate this, we used bisphosphorylated peptides as a substrate and tracked the dephosphorylation of each pTyr using [1]H NMR. Specifically, we measured the appearance of the individual YpY, pYY and YY products over time (Supplementary Fig. S6). As shown in Fig. 6c, as the reaction progresses the concentration of JAK pYY increases more rapidly than YpY, confirming that the second phosphotyrosine is the preferred phosphotyrosine within the JAK2 activation loop. The second pTyr is dephosphorylated at a ~3-fold higher rate than the first. In contrast, when the IRK peptide is used as a substrate there is a slight (~1.6-fold) preference for dephosphorylation of the first pTyr (Fig. 6c). As our structures indicated the involvement of Arg47 in binding the first pTyr outside the catalytic pocket 6we

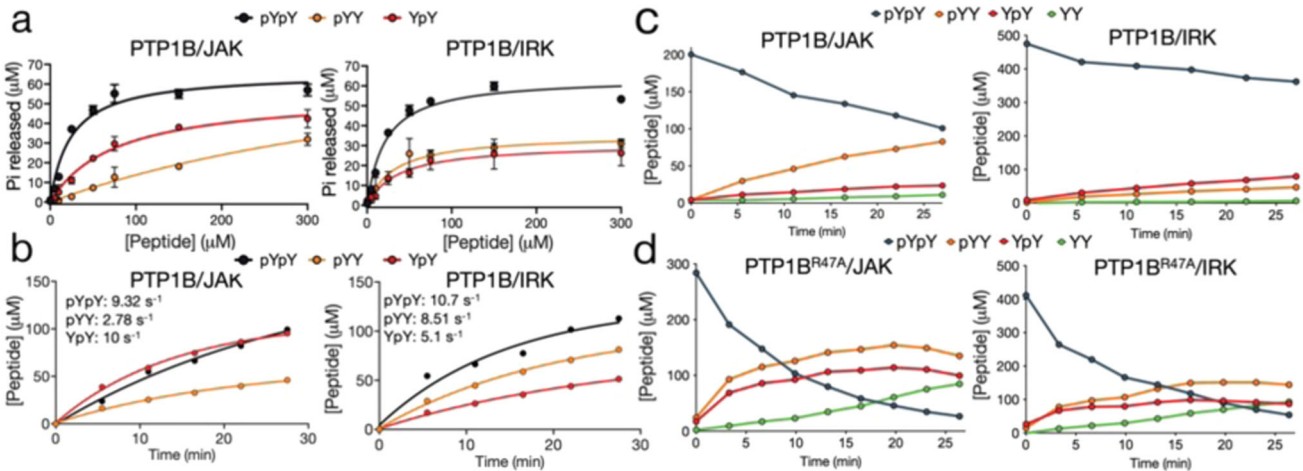

**Fig. 6 PTP1B preferentially targets different phosphotyrosines in the JAK2 and IRK activation loops. a** Michaelis–Menten analysis of PTP1B-induced dephosphorylation via the malachite green assay. Data are from three independent experiments, error bars are SEM. **b** Dephosphorylation of mono- and bisphosphorylated peptides over time as measured by 1H NMR to determine reaction rate (shown as s-1). **c** 1H NMR was used to track the dephosphorylation of JAK2 and IRK bisphosphorylated activation loop peptides over time. PTP1B has a clear preference for the JAK2 downstream phosphotyrosine in contrast to IRK where it displays a mild preference for the upstream phosphotyrosine. **d** Dephosphorylation of JAK2 and IRK bisphosphorylated activation loop peptides over time by PTP1B R47A by 1H NMR. Arg47 has an effect on the preference for dephosphorylation of the second phosphotyrosine in JAK2, with loss of selectivity for dephosphorylation of either phosphotyrosine in the R47A PTP1B mutant. Data are representative of one or two independent experiments.

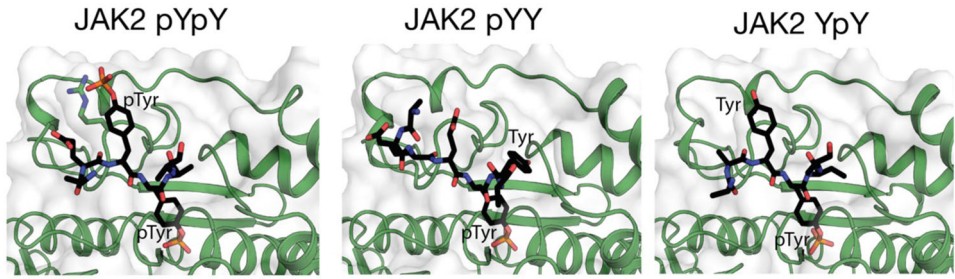

**Fig. 7 PTP1B binds both monophosphorylated forms of the activation loop.** Structures of PTP1B D1818A/Q262A/C215A in complex with pYpY, pYY and YpY JAK2 activation loop peptides. For both monophosphorylated peptides, the phosphorylated tyrosine occupies the catalytic pocket.

tested whether mutation of that arginine affected pTyr preference. As shown in Fig. 5d, mutation of R47 to alanine reduced the preference for dephosphorylation of the second pTyr in JAK2 from 3-fold to 1.3-fold, suggesting that this amino acid plays a role in determining substrate preference.

Given the high sequence similarity between the JAK2 and IRK activation loops, particularly within the residues immediately surrounding the pTyrs (DKE[pY][pY]KVK and ETD[pY][pY]RKG respectively), we were surprised to find PTP1B displayed a preference for different phosphotyrosines in each. To investigate this further, we mutated, one by one, the 2 residues upstream and downstream in the JAK2 sequence to the corresponding residues in IRK. Surprisingly, all four mutant peptides displayed the same preference for dephosphorylation of the second pTyr (Fig. S5). We chose those 4 amino acids as they were the only ones seen to contact PTP1B in our structures. However, it may be that residues outside this region are responsible for driving preference for the second pTyr or it may be that no single amino acid is responsible. Therefore, it remains unclear what characteristics of the JAK sequence cause it to be dephosphorylated differently to IRK.

Interestingly, previous studies of the activity of the JAKs has revealed that the first tyrosine of the JAK2 activation loop is essential for activation of the kinase, whereas the second phosphotyrosine is dispensable[29]. This raises interesting questions as to why the two phosphatases that target JAK show a

preference for the second phosphotyrosine over the first. Further studies of JAK activity revealed that JAK2 is most active when both tyrosine residues are phosphorylated[30]. This may suggest that this preference for the second pTyr in JAK2 has evolved as a means for PTP1B to only dephosphorylate JAK2 when fully activated or that PTP1B is only recruited to dephosphorylate JAK2 when a threshold of activation is reached, i.e. when both tyrosine residues become phosphorylated.

**The catalytic site of PTP1B can accommodate either pTyr in its active site.** Although our structural and biochemical data show that PTP1B exhibits a preference for dephosphorylation of the second pTyr of JAK2, it is clear that both pTyr residues can be dephosphorylated (Fig. 6). In order to observe the first pTyr in the active site, we solved structures of PTP1B (D181A/Q262A/C215A) bound to a substrate peptide in which only the first or second tyrosine was phosphorylated (ie. pYY or YpY) to 2.34 and 3.09 Å resolution respectively (Supplementary Table 1). These structures confirmed that either phosphotyrosine can occupy the catalytic pocket (Fig. 7). Two copies of the PTP1B and peptide are present in the asymmetric unit for the PTP1B pYY structure, with each copy of PTP1B forming different contacts with the JAK2 pYY peptide. In one copy, Arg 47 of PTP1B interacts with the −3 Asp of the peptide, and the carbonyl oxygen of Lys 41 in PTP1B

interacts with the −2 Lys of the peptide. In the other copy, Arg 47 interacts with the −2 Glu (1006) in the peptide. The differences in contacts between PTP1B and the peptide between the two structures may be due to crystal contacts, as in the first copy, the crystal contact likely holds the peptide in place. In both structures, the unphosphorylated tyrosine (Tyr 1008) interacts with Arg 24 and Arg 254 of PTP1B via waters in the second aryl binding site. Three copies of the PTP1B:YpY peptide complex are present in the asymmetric unit, however one was not fully resolvable. In those two copies that were fully resolved, the structure is highly similar to the pYpY structures presented above and also to previously published structure of C215S PTP1B in complex with the peptide ELEFpYMDYE (PDB ID: 1EEO)[17]. This peptide contains a Phe in the −1 position and was shown to be a potent PTP1B substrate.

## Discussion

The use of substrate-trapping mutants has been beneficial in defining cellular targets of PTP1B[7]. The key attribute of an ideal substrate-trapping mutant is that they have negligible, preferably zero, catalytic activity and that they bind to substrates with high affinity. More specifically, the off-rate of the mutant/substrate complex should be low, as this allows isolation of an enzyme/substrate complex that remains stable during immunoprecipitation and multiple washing steps. PTP1B D181A/Q262A has extremely low catalytic activity and has previously been shown to be an excellent substrate-trapping mutant[20]. Although when used in structural experiments we show it to be product-trapping rather than substrate trapping, this is presumably due to the many days in which the enzyme and substrate are co-incubated within the crystal. In biochemical measurements by SPR, performed under conditions where the amount of catalysis that occurs (during the 3 min the enzyme is in contact with the substrate bound to the chip) is negligible, we observed it to bind substrate with high affinity (400 nM) and a reasonably slow off-rate ($0.01 \, \mathrm{s}^{-1}$). However, we discovered that a D181A/Q262A/C215A triple mutant of PTP1B has zero catalytic activity (as expected), much higher affinity (sub-nanomolar) and a thousand-fold reduction in off-rate compared to C215A or D181A/Q262A alone, yielding a complex half-life of >5 h. This off-rate is slow enough to result in the free- and bound-forms of the enzyme appearing as two separate melting curves during the course of a thermal shift assay acquisition.

The reason for the extremely slow off-rate compared to C215A alone is unclear, however we note that incorporation of the D181A/Q262A mutations seem to stabilise the WPD-loop in a closed conformation, in which F182 may restrict exit of the tyrosine from the active site. This preference for the closed conformation can be seen in the structure of the Q262A mutant alone[25]. Likewise, the WPD-loop open conformation is stabilised by an interaction between Arg112 and Asp181 (PDB 2CM2[26]), an interaction that is also not possible in the double or triple mutant. Therefore, we hypothesize that the high affinity between the triple mutant and substrate is due to the already substantial affinity of the catalytically-dead C215A form added to a stabilised WPD-loop down form promoted by the D181A/Q262A mutations, which prevents peptide release. It is hoped that knowledge of the enhanced substrate-trapping properties of this mutant will be useful to others in the field in identifying novel PTP1B substrates.

Here we show that PTP1B overexpression is able to inhibit JAK/STAT signalling by IL6 in the M1 myeloid leukemia cell-line, adding another potential target to the list of cytokines regulated by this phosphatase in vivo. In dephosphorylation assays, the kinase domains of all four JAK proteins could act as substrates for PTP1B dephosphorylation, however, in vivo, it is clear that JAK2 and TYK2 are preferred substrates. Specificity in vivo could be due to regions of either JAK or PTP1B outside their catalytic domains, which were studied in isolation here.

To characterise the interaction between PTP1B and the JAK activation loop we used synthetic peptides of these kinases to solve structures of activation loop motifs bound to PTP1B. These structures indicated a difference between the interaction of PTP1B and the JAK activation loop compared to its interaction with IRK. In both the JAK2 and TYK2 bisphosphorylated activation loop structures (with the catalytically-dead D181A/Q262A/C215A mutant), the second pTyr occupied the catalytic pocket, whereas previous structures with the IRK activation loop showed the opposite. These structures are similar to those of the phosphatase SHP1 in complex with the same substrates[16]. In order to confirm that preference for dephosphorylation of the second pTyr exists in the wild-type protein we used NMR to track the dephosphorylation of each pTyr within the bisphosphorylated substrate. Our results showed a clear preference for the second pTyr in the case of JAK2 but a minor preference for the first pTyr when the IRK activation loop was the substrate. A consistency between the two substrates is that the enzyme favours both of them in their bisphosphorylated, compared to monophosphorylated, forms as previously shown[7].

The location of the second pTyr in the active site means that the second aryl binding site is unoccupied. Instead, the first pTyr is located on a shallow basic surface centered upon R47. An R47A mutant of PTP1B lost its preference for the second pTyr. Given that Arg 47 appears to be important in forming interactions with substates, along with Arg 24 and Arg 264 in the second aryl binding site, we hypothesise that a small molecule inhibitor of PTP1B that engages all three pTyr binding sites may result in increased affinity. Our results highlighted differences in interactions of PTP1B with the IRK and JAK activation loops. However, by necessity our structural and biochemical analyses (as well as previous studies by other groups) used synthetic activation loop peptides as substrates, rather than intact JAK and IRK proteins. The limitations of using a peptide, rather than the entire protein, must be noted and it may be that regions outside the activation loop in the intact protein will drive substrate specificity. Therefore, it is not conclusive that these results can be extrapolated to represent the situation in vivo. A detailed analysis of PTP1B with full-length JAK and IRK proteins is required to gain a full understanding of PTP1B with its substrates and our future efforts lie in this direction.

Taken together, these results expand our current understanding of how phosphatases bind substrates and provide a detailed picture of how the PTP1B phosphatase domain interacts with its main targets, IRK and JAK. It is our hope that these findings will allow for the development of more effective and specific inhibitors against PTP1B to enhance current therapies.

## Methods

**Expression and purification of WT and R47A PTP1B.** DNA encoding human the human PTP1B phosphatase domain (residues 2–321) with a GST tag was cloned into a pGEX vector and transformed into BL21 (DE3) E. coli cells and expression was induced by addition of 1 mM IPTG at 18 °C overnight. Cells were collected by centrifugation and pellets were stored at −30 °C. Cells from 1 L culture were resuspended in 40 mL lysis buffer (20 mM Tris (pH 8.0), 250 mM NaCl, 1 mM PMSF, 2 mM TCEP, and 20 mg lysozyme) and lysed by sonication. Lysate was clarified by spinning cells for 10 min at 20,000 × g before loading supernatant onto GST resin. Bound proteins were washed with 20 mM Tris (pH 8.0), 250 mM NaCl, 2 mM TCEP. Beads were then resuspended in 20 mM Tris (pH 8.0), 300 mM NaCl, 2 mM TCEP with 0.5 mg TEV protease and incubated on a roller at 4 °C overnight. Cleaved PTP1B was separated from the GST tag through a gravity flow column. Eluate was then further purified using size exclusion chromatography (Superdex 200 16/600 from GE healthcare) in TBS, 2 mM TCEP.

**Table 1 Antibodies.**

| Antibody | Species | Dilution | Supplier | Code |
|---|---|---|---|---|
| Primary Antibodies | | | | |
| Phospho-JAK1 | Rabbit | 1:1000 | Santa Cruz Biotech | 44-422 G |
| FLAG | Rat | 1:1000 | WEHI MAB | 9H1 |
| HALO | Mouse | 1:1000 | Promega | G9211 |
| PTP1B | Mouse | 1:1000 | Abcam | ab124375 |
| Actin-HRP | Mouse | 1:10,000 | Santa Cruz | Sc-47778 |
| Secondary Antibodies | | | | |
| Anti rabbit | Donkey | 1:15,000 | GE/Amersham | NA934 |
| Anti mouse | Sheep | 1:15,000 | GE/Amersham | NA931 |
| Anti rat | Donkey | 1:15,000 | ThermoFisher | 31470 |

**Expression and purification DQ and DQC PTP1B.** DNA encoding the WT human PTP1B phosphatase domain (residues 2–321 D181A/Q262/C215A mutants) and an N-terminal His was cloned into a pPROEX vector and transformed into BL21 (DE3) E. coli cells and expression was induced by addition of 1 mM IPTG at 18 °C overnight. Cells were collected by centrifugation and pellets were stored at −30 °C. Cells from 1 L culture were resuspended in 40 mL 20 mM Tris (pH 8.0), 5 mM imidazole, 250 mM NaCl, 1 mM PMSF, 2 mM TCEP, and 20 mg lysozyme and lysed by sonication. Lysate was clarified by centrifugation at 20,000 g before loading supernatant onto 1 mL of Nickel resin. Bound proteins were washed with 20 mM Tris (pH 8.0), 10 mM imidazole (pH 8.0), 250 mM NaCl, and 2 mM TCEP followed by 20 mM Tris (pH 8.0), 30 mM imidazole, 250 mM NaCl, and 2 mM TCEP and eluted in 20 mM Tris (pH 8.0), 250 mM imidazole, 250 mM NaCl, and 2 mM TCEP. Eluate was then further purified using size exclusion chromatography (Superdex 200 16/600 from GE healthcare) in TBS pH 8.0, 2 mM TCEP.

**Malachite green peptide dephosphorylation assay.** All assays were performed using 2 nM WT PTP1B and a Malachite green assay kit (Sigma Aldrich) in TBS, 2 mM TCEP and 0.01 mg/ml BSA in 96-well plates. 80 μL reactions were set up with peptide at varying concentrations and reactions were started by the addition of PTP1B. After incubation times, reactions were quenched by the addition of working reagent provided in the assay kit. Plates were read using a chameleon V plate reader at 620 nm and results were analyzed using Prism[31]. For all assays, two technical replicates were used and a peptide minus PTP1B was measured to determine the amount of free phosphate already in solution, and this was subtracted from the amount measured in the reactions.

**JAK kinase domain dephosphorylation assays.** PTP1B was added to activated JAK kinase domain in 10% (v/v) glycerol, 20 mM Tris (pH 8.0), 500 mM NaCl, 2 mM TCEP and 5 mM EDTA. Samples were taken at each time point required for the assay and added directly to 4x loading dye to stop the reaction. JAK kinase domains were used at 13 μM unless otherwise specified. Reactions all contained JAK kinase domain, 5 mM EDTA, TBS + 0.5% BSA, and PTP1B and those performed with inhibitors contained 5 nM inhibitor or DMSO as a control. Samples were separated by size on 4–15% (w/v) Novex Nupage Tris-Glycine pre-cast gels (Invitrogen). Proteins were separated according to size at 140 V for 1 h in SDS running buffer. Protein was transferred to a methanol activated Millipore PDVF membrane in transfer buffer at 80 V for 2 h or by semi-dry transfer using the iBlot system as per the manufacturer's instructions. Membranes were blocked in TBST containing 5% (w/v) skim milk powder and 0.1% (v/v) Tween-20 at room temperature for 10 min. Primary antibodies were used at dilutions listed in Table 1 in TBST with 5% (w/v) skim milk powder or 5% BSA. Membranes were incubated overnight in 1–5 mL of diluted antibody at 4 °C. Antibodies are listed in Table 1. Membranes were washed 3 times in TBS with 1% (v/v) Tween-20 before probing with HRP conjugated secondary antibody. Each membrane was incubated in 10 mL of diluted secondary antibody for at least 1 h at room temperature. Membranes were washed 6 times for 10 min and exposed to 1 mL of ECL detection reagent (Amersham; Thermofisher scientific) and imaged using a BioRad Chemidoc MP. Image analysis was performed using Image J software.

**Overexpression of proteins in cell lines.** Murine PTP1B with an N-terminal Halo tag was sub-cloned into the Tet-On pfTRE3G expression vector for generation of a stable doxycycline-inducible Halo-mPTP1B expressing HEK293T cell line. HEK293T cells were transfected with TRE3G-Halo-mPTP1B and packaging vectors (psPAX and VSV) using Lipofectamine 2000 (Invitrogen) according to the manufacturer's instructions. Media was replaced 24 h post-transfection and cells were incubated for a further 48 h. Lentivirus was harvested and passed through a 0.45 μM filter prior to infection of target HEK293T cells. Lentivirus was incubated with target cells for 24 h prior to recovery in fresh media for 3 days prior to selection in puromycin (5 ng/mL) for 5 days. To analyse JAK1 phosphorylation,

500 ng Flag-JAK1 was transiently transfected into $0.25 \times 10^6$ doxycycline-treated (1 μg/ml) Halo-mPTP1B expressing HEK293T cells. 24 h post-transfection, cells were lysed in sample reducing buffer and analysed by SDS-PAGE and immunoblot as above (dephosphorylation assays). For M1 cells expressing phosphatases, HEK293T cells were transfected with pCFB_sfi1_LIC vector containing each phosphatase. and packaging vectors (psPAX. and VSV) using Lipofectamine 2000 (Invitrogen) according to the manufacturer's instructions. Media was replaced 24 h post-transfection and cells were incubated for a further 48 h. Lentivirus was harvested and passed through a 0.45 μM filter prior to infection of target HEK293T cells. $5 \times 10^5$ M1 cells/well were seeded in a 24-well plate and 1 mL of lentivirus supernatant was added to a separate well of each plate followed by the addition of 4 mg/ml of polybrene. Cells were spin infected for 1–2 h at 2000 rpm in a Multifuge 2SR with a Swing-out 4 place rotor (Heraeus) at 37 °C. Plates were then incubated overnight at 37 °C in 10% $CO_2$. The following day the cell suspensions were spun at 1500 rpm for 5 min. Cell pellets were resuspended in 1 mL of fresh DMEM-10% and seeded in a new 24-well plate. Following incubation overnight at 37 °C in 10% $CO_2$, cell suspensions were transferred to a 12-well plate and 1 mL of DMEM-10%. For cells infected with lentivirus, DMEM-10% supplemented with puromycin was used to select for those cells the virus had infected. Cells were then incubated for 3 d at 37 °C in 10% $CO_2$.

**NMR dephosphorylation analysis of activation loop peptides.** 0.5 mg of JAK2 or IRK pYpY, pYY and YpY peptides were dissolved in 500 μl of 20 mM Tris (pH 7.5), 2 mM TCEP, and 100 mM NaCl. Dissolved peptide was then added to 25 μL D2O. NMR spectra were collected using a Bruker Advance 600 MHz or 800 MHz spectrometer. To start the dephosphorylation reaction, 2 nM PTP1B was added to each sample and multiple 1H NMR spectra were recorded over the course of 1 h. Data was analyzed using TopSpin v3.2 (Bruker). The concentration of each species of ALP (pYpY, pYY, or YpY) in the reaction was calculated by integrating the area under the peak and normalising these values to the starting peptide concentration.

**Thermostability assays.** For peptide and phenyl phosphate experiments, proteins were desalted into 100 mM NaCl, 20 mM Tris (pH 8.0), 2 mM TCEP buffer and diluted to 100 μM. Where peptides were used, a five-fold molar excess of peptide was added to each sample, in phenyl phosphate conditions its concentration was 8 mM. For experiments with phosphotyrosine, proteins were diluted to 30 μM in PBS and in its concentration was 10 mM. 10 uL of each sample was transferred into a capillary and measured from 35 to 95 °C using a Tycho NT.6 (Nanotemper). Data were analyzed in Prism.

**SPR analysis**

*Single-cycle kinetics.* Biotinylated JAK activation loop peptide was coupled to a CAP chip using the CAPture system (Cytiva) and run on a Biacore 8k instrument. PTP1B was flowed over the surface of the chip at 30 ul/min using the single-cycle kinetics protocol with 2 min association and a final 30-minute dissociation step. The top concentration of the WT, D181A/Q262A and C215A mutants was 2 uM with 4 subsequent 2-fold dilutions. The top concentration of the D181A/Q262A/ C215A triple mutant was 125 nM with 4 subsequent 2-fold dilutions. A blank series was included for subtraction. Data was analysed using the Insight Evaluation software and fit to a 1:1 binding model without drift or bulk refractive index change.

*Multi-cycle kinetics.* Biotinylated JAK activation loop peptide was coupled to an SA chip using and run on a Biacore 4000 instrument. PTP1B was flowed over the surface of the chip at 30ul/min using a multi-cycle kinetics protocol with 7 min association and a 2 min dissociation steps. The chip was regenerated with 50 mM NaOH/1 M NaCl after each injection. The top concentration of all PTP1B mutants was 1 μM with 4 subsequent 2-fold dilutions. A blank cycle was included for subtraction. Data was analysed using the biaevaluation software and fit to a 1:1 binding model without drift or bulk refractive index change.

**JAK kinase domains expression and purification.** Sf21 insect cells were seeded into a fernbach flask at $2.5–3.0 \times 10^6$ cells/mL and infected with baculovirus at a multiplicity of infection of approximately 3.0. Cells were incubated at 27 °C for 48 h and subsequently harvested by centrifugation at 5000 g for 30 min and pellets were stored at −30 °C until purification. For purification, cells were resuspended and lysed in 10 mM Tris (pH 7.5), 150 mM NaCl, 2 mM TCEP, 1 mM PMSF, 5 U/mL DNAseI, by sonication on an ice bath. Lysate was clarified by spinning cells for 60 min at 40,000 g before being filtered through a 0.8 μm disposable filter unit. Filtered supernatant was loaded onto a HisTrap cartridge and washed with 20 mL 2% of Nickel buffer (20% (v/v) glycerol, 20 mM Tris (pH 8.0), 500 mM NaCl, 5 mM imidazole and 2 mM TCEP), and subsequently 15 mL of 7% Nickel buffer (20% (v/v) glycerol, 20 mM Tris (pH 8.0), 500 mM NaCl, 5 mM imidazole and 2 mM TCEP). Full length JAK1 was eluted in 3 mL 20% (v/v) glycerol, 20 mM Tris (pH 8.0), 500 mM NaCl, 500 mM imidazole and 2 mM TCEP. Eluate was then further purified using size exclusion chromatography (Superdex 200 16/600 from GE healthcare) in 10% glycerol, 20 mM Tris (pH 8.0), 500 mM NaCl and 2 mM TCEP.

**Table 2 Peptide sequences.**

| Name | Sequence | MW |
|---|---|---|
| JAK1 pYpY | AIETDKE[pY][pY]TVKDDLD | 2077 |
| JAK2 pYpY | VLPQDKE[pY][pY]KVKEPGE | 2002 |
| JAK3 pYpY | LLPLDKD[pY][pY]VVREPGQ | 1823 |
| TYK2 pYpY | VPEGHE[pY][pY]RVREDGD | 1980 |
| JAK2 pYY | VLPQDKE[pY]YKVKEPGE | 2001 |
| JAK2 YpY | VLPQDKEY[pY]KVKEPGE | 2001 |
| JAK2 Ala | VLPQDKEA[pY]KVKEPGE | 1780 |
| IRK pYpY | ETD[pY][pY]RKGGKGL | 1658 |
| IRK pYY | ETD[pY]YRKGGKGL | 1578 |
| IRK YpY | ETDY[pY]RKGGKGL | 1578 |
| JAK3 pY785 | ISSD[pY]ELLSDPT | 1418 |
| JAK2 pYpY KT | VLPQDTE[pY][pY]KVKEPGE | 2152 |
| JAK2 pYpY ED | VLPQDKD[pY][pY]KVKEPGE | 2151 |
| JAK2 pYpY VK | VLPQDKE[pY][pY]RVKEPGE | 2152 |
| JAK2 pYpY KR | VLPQDKE[pY][pY] KKKEPGE | 2151 |

**Crystallography**. All proteins were buffer exchanged into low salt (20 mM Tris (pH 7.5), 100 mM NaCl, 2 mM TCEP) and crystal trays were set up at 10 or 20 mg/mL of protein with a 2-fold excess of peptide using vapour diffusion sitting drop experiments at the collaborative crystallisation centre, CSIRO, or hanging drop set up in house. Conditions for each structure are listed below. All crystals were cryo-protected in mother liquor supplemented with either 20% or 25% ethylene glycol and immediately snap frozen in liquid nitrogen. Data was collected at the MX2 beamline and the Australian synchrotron[32]. Data reduction, scaling and integration was performed using XDS. Each structure was solved by molecular replacement (search model PDBID: 1SUG or 4ZRT) using Phaser as implemented in PHENIX[33].

**Refinement of structures with peptide**. All structures were refined using PHENIX[33] and model building was performed in COOT[34]. To prevent bias of maps, we followed a specific strategy when building peptides into our structures. In summary, for the initial phases of refinement, no peptide was included. Upon clear density of peptide becoming apparent, initially, only a pTyr was placed in the catalytic pocket for refinement followed by the peptide backbone. Only when it was clear as to the register of the peptide were the second pTyr and other residues built. For structures where this was not clear (D/Q) both registers were refined. Composite OMIT maps were generated using simulated annealing as implemented in PHENIX[33].

**Protein and peptide crystallisation conditions**. All peptides including sequences used are listed in Table 2. Below are the crystallisation conditions for each structure presented.

PTP1B D/Q Apo 12% PEG 8000, 0.15 M Mg Acetate, 0.1 M MES (pH 6.5)

PTP1B D/Q JAK1 pYpY 12% Peg 4000, 0.1 M Calcium acetate, 0.05 M MES (pH 6.5)

PTP1B D/Q JAK2 pYpY 14% PEG 8000, 0.10 M Mg Acetate, 0.1 M MES (pH 6.5)

PTP1B D/Q JAK3 pYpY 12% Peg 4000, 0.15 M Calcium acetate, 0.05 M MES (pH 6.5)

PTP1B D/Q TYK2 pYpY 0.2 M Ca Acetate, 12.5% PEG 4000, 0.05 M MES (pH 6.5)

PTP1B D/Q/C JAK2 pYpY 0.2 M Mg formate, 20% PEG 3350

PTP1B D/Q/C JAK2 pYY 14% PEG 8000, 0.20 M Mg Acetate, 0.1 M MES (pH 6.5)

PTP1B D/Q/C JAK2 YpY 25% w/v PEG 3350, 0.2 M NaCl, 0.1 M Tris Cl (pH 8.5)

PTP1B D/Q/C TYK2 pYpY 14% PEG 8000, 0.20 M Mg Acetate, 0.1 M MES (pH 6.5)

**Statistics and reproducibility**. Incucyte assay were performed independently 3 times and analysed using Essenbiosciences Incucyte software. Western blots were independently repeated between 3 and 4 times each. Biacore experiments were performed up to 5 times and independently at least twice then analysed using insight evaluation software and prism software. Malachite green assays were performed individually three times and analysed using prism software. NMR experiments were performed once or twice as indicated and analysed using Top-Spin software. Spread of data are shown throughout paper through use of both individual datapoints and SD or SEM error bars as indicated.

**Reporting summary**. Further information on research design is available in the Nature Portfolio Reporting Summary linked to this article.

## Data availability

Data that support this study are available from the corresponding author upon reasonable request. Atomic coordinates for all structures have been deposited in the Protein Data Bank with the following accession numbers: PTP1B DQ/JAK1 (PDB ID: 8EXJ), PTP1B DQ/JAK2 (PDB ID: 8EXK), PTP1B DQ/JAK3 (PDB ID: 8EXM), PTP1B DQ/TYK2 (PDB ID: 8EXN), PTP1B DQ apo (PDB ID: 8EXI), PTP1B DQC/JAK2 pYpY (PDB ID: 8EYB), PTP1B DQC/TYK2 pYpY (PDB ID: 8EYC), PTP1B DQC/JAK2 pYY (PDB ID: 8EYA) AND PTP1B DQC/JAK2 YpY (PDB ID: 8F88). Source data are provided with this paper including all raw blots (Supplementary Fig. 7 and Supplementary Data 1–7).

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

## Acknowledgements

This work was supported by the National Health and Medical Research Council (NHMRC) Australia (Project grant no. 1122999, Program grant no. 1113577), an NHMRC IRIISS Grant 9000220, and a Victorian State Government Operational Infrastructure Scheme grant. J.J.B. is supported by an NHMRC fellowship. RM was supported by an Australian Postgraduate Award. This research was undertaken in part using the MX2 beamline at the Australian Synchrotron, part of ANSTO, and made use of the Australian Cancer Research Foundation (ACRF) Eiger detector.

## Author contributions

R.M., N.K., C.T., H.C., A.L., T.S., N.A.N., and N.P.D.L. carried out experiments. T.T. provided several protein constructs. R.M., N.K., and J.J.B. designed experiments and analysed and interpreted data. R.M., N.J.K., collected and analysed crystallographic data. R.M., N.J.K., J.J.B. wrote the initial manuscript. All authors revised the manuscript.

## Competing interests

The authors declare no competing interests.
