## [Peer Review File · Communications Biology]

Reviewers' comments:

Reviewer #1 (Remarks to the Author):

In PTP1B-IRK interaction, PTP1B dephosphorylate the N-terminal pY first while the pY+1 interacts with a positively charged patch on the surface of PTP1B referred to as the second aryl binding site. Here the authors claimed that they identified a new catalysis pattern of PTP1B dephosphorylating bisphosphorylated (X-pY-pY-X) JAK, which is different from previous reported PTP1B-IRK interaction. This specific new pattern is that the C-terminal phosphotyrosine is preferred in PTP1B-JAK interaction, and R47 of PTP1B is essential in this specific pattern. This is the main novelty of this study, but this conclusion seems to have a gap with the performed experiments.

1. The authors claimed that R47 of PTP1B, but not the substrate sequence, determines the preferred phosphotyrosine, what results in the difference between JAK and IRK?
2. The conclusion that the C-terminal phosphotyrosine is preferred in PTP1B-JAK interaction should be further validated by more convincing experiments. NMR experiment is not satisfactory and the structure analysis of PTP1BR47A-JAK is needed.

Some minor issues should be corrected.

1. The authors said experiments are in triplicate, but it is hard to tell because there is no error bar.
2. The quality of several images is too low. For example, fig 2c and 3d, I can not distinguish subgroups because the colors and locations are too similar.
3. The results of experiments or figure legend should be more detailed. For example. What assay was performed in figure 1c? what are the different treatments? Why p-JAK1 has a molecular weight about 37kd, and why left and right panel has different molecular weight of p-JAK1
4. some typos should be corrected. For example, there are two "using" in line 307.

Reviewer #2 (Remarks to the Author):

The manuscript by Morris et al. describes the novel interaction mechanism between PTP1B and JAK. Unlike the PTP1B-IRK interaction, the authors revealed that the PTP1B active site interacted with the second phosphotyrosine of JAK. The authors presented biochemical and structural data to support the interaction mechanism. While the biochemical data seem relevant, the structural data need to be more convincing. Specific comments are as follows.

1. Fig. 4 shows electron density for only the vicinity of phosphotyrosines. From this electron density map, one cannot conclude that the second phosphotyrosine is bound to the active site. The proceeding direction of the peptide could be the opposite. Thus, the authors must show electron density of more residues of the peptide and specific interactions of the peptide-PTP1B in the more distant residues have to be described.
2. Although the authors indicated that electron density maps of supplementary figures are SA-omit maps, in Fig. 4, the method for the omit map generation is lacking. If Fig. 4 did not use the SA-omit map protocol, the electron density credibility is low. Thus, the authors must specify the omit map protocol in Fig. 4.
3. About the 2/3 of the discussion section elaborates the substrate-trapping mutants, which dilutes the main points of this manuscript. The substrate-trapping parts should be moved to the results section with minimizing explanation. And the discussion section has to be focused on the interpretation and implication of the PTP1B-JAK interaction data.
4. From the number of protein atoms, it appears that some structures presented in the manuscript have 2-3 protein molecules in the asymmetric unit. The authors must comment on the differences of the multiple molecules in the asymmetric unit. Furthermore, the multiple protein molecules in the asymmetric unit appear to have incompletely-modeled parts, which has to be commented in the manuscript.

(Typos)

1. Line 349-350, PDB ID is needed.
2. Fig. 4b, The black chain number is blue and the blue chain number is black. It is preferable to match the colors.
3. Line 75, PTP1b ->PTP1B
4. Line 307, Using using -> twice written
5. Line 452, R47, Arg 47 are mixed. One letter or three letter has to be used consistently.
6. Line 480, mLn20 mM
7. Line 498, 20 mM Tris. -> extra period
8. Line 510, 4C
9. Line 537-557, the space between number and unit is inconsistent.
10. Line 552, 558, the parenthesis of pH is inconsistent.
11. Line 556-557, Specify the "Nickel Buffer"
12. Line 568, All structurers -> misspelled

Reviewer #3 (Remarks to the Author):

The manuscript by Morris, et. al. ("Structure guided studies of the interaction between PTP1B and JAK") presents a study of Protein Tyrosine Phosphatase 1B (PTP1B) interactions with Janus Kinases (JAKs). JAK dephosphorylation by PTP1B is implicated in the regulation of the JAK-STAT signaling pathway. Here, the authors perform biochemical and structural studies to clarify the molecular basis for the PTP1B-JAK interaction. An important observation is that PTP1B preferentially targets the second phosphorylation site on the JAK activation loop, which is different from that previously reported for a closely related PTP1B substrate, insulin receptor kinase (IRK). The experiments are well-presented and appear to have been performed appropriately. The data support the overall conclusions, with the exception of the comment below. This work will be of interest to the field.

The following points should be addressed by the authors:

Important details for the JAK substrates used in the SPR and thermal shift experiments (Fig 2) are not fully described. What is the sequence, length, etc. of the JAK3 activation loop that was immobilized to the SPR chip (referred to on line 162)? Similar details are missing for the JAK1 phosphopeptide described on line 170. What is the chemical structure of the non-hydrolyzable phosphotyrosine analog (JAK(diF))?

The refinement of the PTP1B DQC/JAK2 pYY crystal structure (PDB ID: 8EYA) appears to be incomplete based on the R/Rfree values (31%/30%) reported in supplementary Table 1. The structure validation report for this structure gives more reasonable R factors. Please check and correct as needed.

In the discussion, it is noted that "...to confirm that preference for dephosphorylation of the second pTyr exists in the wild-type protein we used NMR to track the dephosphorylation of each pTyr within the bisphosphorylated substrate" (lines 443-446). It should be clarified that while this experiment showed preference for the second pTyr in the peptide substrates, it does not confirm this preference in the full-length wild-type protein. The authors do hint at this in line 435, but the wording of the sentence cited above could be somewhat misleading. Is it possible that the observed difference in JAK binding compared to IRK binding in the crystal structures is partly a function of crystallizing with short peptides rather than the full-length protein? A discussion of this possibility should be included.

Minor comments:

Ln 104 - Italicize *in vitro*

Ln 147 - "hydrolyses" should be "hydrolyzes"

Ln 223 - "...with Phe182 is partially..." delete "is"

Ln 319 - "then" should be "the"

Ln 322 - "effected" should be "affected"

Ln 475 - "DNA encoding human the WT human..." delete the first "human"

Ln 477 - "clone" should be "cloned"

Ln 480 - "mLn20 mM Tris" something needs to be corrected

We thank the reviewers for their very careful analysis and critique of our work. We believe we have addressed all of their concerns with the exception of one (which we will argue is outside the scope of this study-this is highlighted in red text). Please see below for a point-by-point response to all concerns.

Reviewer #1 (Remarks to the Author):

1. The authors claimed that R47 of PTP1B, but not the substrate sequence, determines the preferred phosphotyrosine, what results in the difference between JAK and IRK?

We thank reviewer one for this question as it highlights the fact that we must not have been clear with our writing. Certainly, it is the sequence of the *substrate* that determines which pTyr is the preferred target (or else both JAK and IRK peptides would show the same pattern). However, we are suggesting that R47 helps co-ordinate the first pTyr when the second is in the active site and therefore that its mutation forces the preference more towards the first pTyr.

In order to try and answer the reviewer's question, we obtained four new phosphopeptides and again used NMR to track which pTyr was dephosphorylated most rapidly. The peptides replaced single residues from JAK1 with those of IRK and were:

Original sequences

JAK: DKE [pY] [pY] KVKEPGE
IRK: ETD [pY] [pY] RKGKGLL

New peptides

JAK (ED) : DK**D** [pY] [pY] KVKEPGE
JAK (KT) : D**T**E [pY] [pY] KVKEPGE
JAK (KR) : DKE [pY] [pY] **R**VKEPGE
JAK (VK) : DKE [pY] [pY] **K**KKEPGE

We chose these peptides as in our structures we only observe electron density for 2 residues on either side of the pTyr and so assumed that these residues would encode pTyr preference. However, all of these new peptides also showed preferential dephosphorylation of the second pTyr, just like the original JAK sequence (See New Supp. Fig. S5 – shown on the next page), but unlike IRK.

We assume that it is not a single residue in the substrate that drives preference for the downstream pTyr. Unfortunately, these doubly-phosphorylated peptides are expensive to obtain and our attempts to synthesize them ourselves have been unsuccessful. We have added the following section to the text.

“Given the high sequence similarity between the JAK2 and IRK activation loops, particularly within the residues immediately surrounding the pTyrs (DKE[pY][pY]KVK

and ETD[pY][pY]RKG respectively), we were surprised to find PTP1B displayed a preference for different phosphotyrosines in each. To investigate this further, we mutated, one by one, the 2 residues upstream and downstream in the JAK2 sequence to the corresponding residues in IRK. Surprisingly, all four mutant peptides displayed the same preference for dephosphorylation of the second pTyr (Figure S5). We chose those 4 amino acids as they were the only ones seen to contact PTP1B in our structures. However, it may be that residues outside this region are responsible for driving preference for the second pTyr or it may be that no single amino acid is responsible. Therefore, it remains unclear what characteristics of the JAK sequence cause it to be dephosphorylated differently to IRK.”

2. The conclusion that the C-terminal phosphotyrosine is preferred in PTP1B-JAK interaction should be further validated by more convincing experiments. NMR experiment is not satisfactory and the structure analysis of PTP1BR47A-JAK is needed.

We politely disagree with the reviewer here. There is no more convincing technique than NMR for this purpose. NMR is the gold standard technique when determining which pTyr is preferred and we are not the first to do this for this purpose, Lee et al have previously published the same protocol for LAR phosphatase (Lee *et al.*, *Protein Science* (1992), 1, 1353-1362.

The reason why NMR is so powerful here is that it can identify all four species (pYpY, pYY, YpY and YY) unambiguously. See Supp Figure 4 for the unique fingerprint of each species. Over time, dephosphorylation of the pYpY form can be tracked as shown here:

As shown, in the early timepoints peaks corresponding to pY-Y (dephosphorylation of the second pTyr, black arrows) appear and increase in magnitude over time much faster than the Y-pY peaks (red arrows). This allows unequivocal identification of the second pTyr as the preferred substrate in this peptide.

In contrast, there are no antibodies available that can distinguish between the different phosphorylated species. Even antibodies that claim to only detect the bisphosphorylated form give a positive signal for monophosphorylated forms.

The reviewer suggest we solve the structure of the R47A form of the protein, however we would need to do so on the background of the DQC mutations (substrate trapping mutant) rather than the wild-type protein. This may not reflect the substrate preference of the natural enzyme which is why we performed the NMR studies in the first place. Finally, our NMR studies show that the preference for pTyr2 is still present when R47 is mutated, it is just reduced. Hence, we would likely observe either the same structure as before, or else one with a mixture of pY1 and pY2 in the active site which would be difficult to

disentangle. We already present 10 different structures in this manuscript and suggest that the extra time taken to re-clone, express, purify and solve the protein structure is outside the scope of this study given that the results cannot be unambiguous.

Some minor issues should be corrected.

1. The authors said experiments are in triplicate, but it is hard to tell because there is no error bar.

We thank reviewer 1 for this helpful comment. Where data have been averaged, error bars have been included and described, however where data is representative of 3 independent experiments, and thus not averaged this was stated in the figure legend and no error bars were shown. We will also provided the raw data for all experiments included.

2. The quality of several images is too low. For example, fig 2c and 3d, I can not distinguish subgroups because the colours and locations are too similar.

We have changed the colours in figure 2 to be more vibrant and hopefully clear to the reader. Figure 3d however we had each protein in the same colour to show that each was highly similar to one another and they are almost indistinguishable, and have left this in its original form.

3. The results of experiments or figure legend should be more detailed. For example. What assay was performed in figure 1c? what are the different treatments? Why p-JAK1 has a molecular weight about 37kd, and why left and right panel has different molecular weight of p-JAK1.

We thank reviewer 1 for this feedback. The figure legends have been updated to include more detailed information on the methods.

To explain the difference in the p-JAK molecular weight, this is due to the use of different recombinant JAKs which do indeed have different molecular weights. P-JAK1 is the antibody used to blot for all of them, as it recognises all activation loops when phosphorylated. To make this clearer, the figure and legend have been updated to indicate this.

4. some typos should be corrected. For example, there are two “using” in line 307.

Typos have been corrected where identified.

Reviewer #2 (Remarks to the Author):

The manuscript by Morris et al. describes the novel interaction mechanism between PTP1B and JAK. Unlike the PTP1B-IRK interaction, the authors revealed that the PTP1B active site interacted with the second phosphotyrosine of JAK. The authors presented biochemical and structural data to support the interaction mechanism. While the biochemical data seem relevant, the structural data need to be more convincing. Specific comments are as follows.

1. Fig. 4 shows electron density for only the vicinity of phosphotyrosines. From this electron density map, one cannot conclude that the second phosphotyrosine is bound to the active site. The proceeding direction of the peptide could be the opposite. Thus, the authors must show electron density of more residues of the peptide and specific interactions of the peptide-PTP1B in the more distant residues have to be described.

(Note that this refers to what is now Fig.5 in the revised edition).

We thank the reviewer for this comment. We were very careful to avoid any bias when building into the electron density and to make sure we had the peptide in the correct register. A summary of our refinement strategy is given here:

We used 4ZRT as the MR model for this structure with the singly phosphorylated pTyr peptide removed. In the initial refinement steps, we deliberately did not build peptide into the difference density until most other aspects had been refined, however it was clear that a pTyr was in the catalytic pocket. A simulated annealing step was used in refinement early in refinement to further reduce bias.

We continued to refine the structure for several rounds without the addition of the peptide, at which stage it became clear only a pTyr would fit the density. Moreover, even at this stage the orientation of the peptide was clear as the pTyr carboxylate was obviously to the right, rather than the left (in the orientation of the figure below).

Therefore we placed a pTyr in the pocket catalytic pocket and stubbed residues on either side to ensure we didn't incorrectly assume a particular register. At that point, from the subsequent difference density it was clear that the pTyr must be in the -1 position.

For clarity, below we show that there is no difference density in the second pTyr pocket which we would expect if another pTyr was positioned in this pocket.

Upon final refinement we could resolve 6 residues of the peptide total. Two on either side of the pTyr-pTyr. We have included a summary of this methodology in the methods section.

Finally, if the reviewer is concerned that the peptide may have been running backwards (C to N, rather than N to C) – at this resolution (2.09 Å) there was clear density for the peptide backbone carbonyls (as shown below) which allowed unambiguous determination of its position and direction. Interactions between the peptide backbone and protein are observed that support the orientation, namely D1161 backbone carbonyl of peptide H-bonds with backbone NH of R47 of PTP1B, and D48 side chain of PTP1B coordinates to the NH of both pY1163 and K1164.

2. Although the authors indicated that electron density maps of supplementary figures are SA-omit maps, in Fig. 4, the method for the omit map generation is lacking. If Fig. 4 did not use the SA-omit map protocol, the electron density credibility is low. Thus, the authors must specify the omit map protocol in Fig. 4.

Apologies – this was performed using the SA-omit map protocol. The details have now been added to the figure legend and as a small addition to the methods section.

3. About the 2/3 of the discussion section elaborates the substrate-trapping mutants, which dilutes the main points of this manuscript. The substrate-trapping parts should be moved to the results section with minimizing explanation. And the discussion section has to be focused on the interpretation and implication of the PTP1B-JAK interaction data.

We feel that the identification of such a powerful substrate-trapping mutant as the DQC triple mutation, and also the characterisation of the DQ double mutant as a “product-trapping” mutant will be of interest to the field. Nevertheless, we have shorted the discussion section as requested, to include only those salient points.

We have also included a concluding statement discussing the limitations of using peptides as surrogates for the whole protein in these types of analyses – a limitation not only of our study but also all the structural and biochemical studies performed previously.

“Our results highlighted differences in interactions of PTP1B with the IRK and JAK activation loops. However, by necessity our structural and biochemical analyses (as well as previous studies by other groups) used synthetic activation loop peptides as substrates, rather than intact JAK and IRK proteins. The limitations of using a peptide, rather than the entire protein, must be noted and it may be that regions outside the activation loop in the

intact protein will drive substrate specificity. Therefore, it is not conclusive that these results can be extrapolated to represent the situation *in vivo*. A detailed analysis of PTP1B with full-length JAK and IRK proteins is required to gain a full understanding of PTP1B with its substrates and our future efforts lie in this direction.”

4. From the number of protein atoms, it appears that some structures presented in the manuscript have 2-3 protein molecules in the asymmetric unit. The authors must comment on the differences of the multiple molecules in the asymmetric unit. Furthermore, the multiple protein molecules in the asymmetric unit appear to have incompletely-modelled parts, which has to be commented in the manuscript.

We thank reviewer 2 for this comments, and have included the following text from line 456 onwards to discuss these structures in more detail.

“Two copies of the PTP1B and peptide are present in the asymmetric unit for the PTP1B pYY structure, with each copy of PTP1B forming different contacts with the JAK2 pYY peptide. In one copy, Arg 47 of PTP1B interacts with the -3 Asp of the peptide, and the carbonyl oxygen of Lys 41 in PTP1B interacts with the -2 Lys of the peptide. In the other copy, Arg 47 interacts with the -2 Glu (1006) in the peptide. The differences in contacts between PTP1B and the -2/-3 positions of the peptide between the two structures appear to be due to the crystal packing. In both structures, the unphosphorylated tyrosine (Tyr 1008) interacts with Arg 24 and Arg 254 of PTP1B via waters in the second aryl binding site.

Three copies of the PTP1B:YpY peptide complex are present in the asymmetric unit, however for one the density was incomplete with large regions missing. In the two copies of PTP1B that were well resolved, the structure is highly similar to the pYpY structures presented above and also to previously published structure of C215S PTP1B in complex with the peptide ELEFpYMDYE (PDB ID: 1EEO) [17]. This peptide contains a Phe in the -1 position and was shown to be a potent PTP1B substrate.”

And “In contrast to IRK, we found that the first phosphotyrosine (phosphotyrosine 1007 of JAK2 and phosphotyrosine 1045 of TYK2) in both of these structures was located in a shallow, basic surface near Arg 47. The guanidinium group of Arg 47 forms a salt bridge with the phosphate while the backbones of both amino-acids form hydrogen bonds with one another. In one copy of the PTP1B:JAK2 complex, the density for the guanidinium group of Arg 47 is weak, which suggests some flexibility of this residue. In all models, the sidechain of Asp 48 formed a hydrogen-bond with the backbone amide of the phosphotyrosine in the catalytic site as well and the +1 residue of the peptide (Supplementary Figure Sx2). These data differ from the previous observation that PTP1B has a preference for the first phosphotyrosine in the activation loop of IRK [7] (Figure 5a) and indicates that PTP1B may interact with different substrates via different modes.”

1. Line 349-350, PDB ID is needed. Added at line 350
2. Fig. 4b, The black chain number is blue and the blue chain number is black. It is preferable to match the colors. Colours of the text for chains has been changed.
3. Line 75, PTP1b ->PTP1B corrected “Here we show that all four JAK kinase domains are dephosphorylated by PTP1B and structurally characterise this interaction using X-ray crystallography.”
4. Line 307, Using using -> twice written deleted “Using ¹H NMR we then measured the dephosphorylation of individual YpY, pYY and YY peptides over time, and similarly, observed an overall preference for bisphosphorylated peptides, however JAK2 YpY was dephosphorylated at a similar rate as the pYpY peptides, highlighting again the preference for this residue by PTP1B”
5. Line 452, R47, Arg 47 are mixed. One letter or three letter has to be used consistently. Changed “The location of the second pTyr in the active site means that the second aryl binding site is unoccupied. Instead, the first pTyr is located on a shallow basic surface centered upon Arg 47. An R47A mutant of PTP1B lost its preference for the second pTyr. Given that Arg 47”
6. Line 480, mLn20 mM
n was deleted, should have been a space “Cells from 1L culture were resuspended in 40 mL 20 mM Tris (pH 8.0), 5 mM imidazole, 250 mM NaCl, 1 mM PMSF, 2 mM TCEP, and 20 mg lysozyme and lysed by sonication.”
7. Line 498, 20 mM Tris. -> extra period removed “. PTP1B was added to activated JAK kinase domain in 10% (v/v) glycerol, 20 mM Tris (pH 8.0), 500 mM NaCl, 2 mM TCEP and 5 mM EDTA.”
8. Line 510, 4C
degrees symbol added “Membranes were incubated overnight in 1-5 mL of diluted antibody at 4°C.”
9. Line 537-557, the space between number and unit is inconsistent.
Spaces added
10. Line 552, 558, the parenthesis of pH is inconsistent.
Brackets added
11. Line 556-557, Specify the “Nickel Buffer”
Specified “with 20 mL 2% of Nickel buffer (20% (v/v) glycerol, 20 mM Tris (pH 8.0), 500 mM NaCl, 5 mM imidazole and 2 mM TCEP), and subsequently 15mL of 7% Nickel buffer (20% (v/v) glycerol, 20 mM Tris (pH 8.0), 500 mM NaCl, 5 mM imidazole and 2 mM TCEP). Full length JAK1 was eluted in 3 mL 20% (v/v) glycerol, 20 mM Tris (pH 8.0), 500 mM NaCl, 500 mM imidazole and 2 mM TCEP.”
12. Line 568, All structurers -> misspelled corrected “All structures were refined using PHENIX [31] and model building was performed in COOT”

Reviewer #3 (Remarks to the Author):

The manuscript by Morris, et. al. (“Structure guided studies of the interaction between

PTP1B and JAK”) presents a study of Protein Tyrosine Phosphatase 1B (PTP1B) interactions with Janus Kinases (JAKs). JAK dephosphorylation by PTP1B is implicated in the regulation of the JAK-STAT signaling pathway. Here, the authors perform biochemical and structural studies to clarify the molecular basis for the PTP1B-JAK interaction. An important observation is that PTP1B preferentially targets the second phosphorylation site on the JAK activation loop, which is different from that previously reported for a closely related PTP1B substrate, insulin receptor kinase (IRK). The experiments are well-presented and appear to have been performed appropriately. The data support the overall conclusions, with the exception of the comment below. This work will be of interest to the field.

The following points should be addressed by the authors:

Important details for the JAK substrates used in the SPR and thermal shift experiments (Fig 2) are not fully described. What is the sequence, length, etc. of the JAK3 activation loop that was immobilized to the SPR chip (referred to on line 162)? Similar details are missing for the JAK1 phosphopeptide described on line 170. What is the chemical structure of the non-hydrolyzable phosphotyrosine analog (JAK(diF))?

We thank reviewer 3 for this valuable feedback. We have added a table with the sequences of all the peptides used in this study to materials and methods and updated the figure legend.

“(A) SPR analysis of PTP1B mutants binding to immobilised JAK3 pY785 activation loop pTyr peptide. The D181A/Q262A/C215A mutant displays a significantly slower off-rate. Data are representative of 3-4 independent experiments. (B) Quantification of the off-rate of the PTP1B/JAK3 pY785 peptide interaction using single-cycle kinetics. The raw data is shown as solid lines with the fitted curves shown as dashed lines. The off-rate of the triple mutant is 1000x slower than the C215A or D181A/Q262A mutants or WT. (C) Analysis of the PTP1B/JAK3 pY785 interaction by thermal shift assay. A non-hydrolyzable form of the peptide was used that contains difluorophosphotyrosine (JAK(diF)). The thermal stability of the WT protein increases gradually with the addition of peptide as expected. In contrast, the D181A/Q262A/C215A mutant data indicates a free- and bound-form that do not interconvert during data acquisition. The raw data is shown in the upper panels and the first derivative of that data in the lower panels. Data are representative of 2 or 3 independent experiments.”

We have also added a figure of the structure of the diF peptide to the supplementary information as shown below:

The refinement of the PTP1B DQC/JAK2 pYY crystal structure (PDB ID: 8EYA) appears to be incomplete based on the R/Rfree values (31%/30%) reported in supplementary Table 1. The structure validation report for this structure gives more reasonable R factors. Please check and correct as needed.

Apologies! Thank you for picking this up – those numbers were incorrect. The table has now been corrected. R free 20.62 (29.57) and R work 24.16 (32.79).

In the discussion, it is noted that “...to confirm that preference for dephosphorylation of the second pTyr exists in the wild-type protein we used NMR to track the dephosphorylation of each pTyr within the bisphosphorylated substrate” (lines 443-446). It should be clarified that while this experiment showed preference for the second pTyr in the peptide substrates, it does not confirm this preference in the full-length wild-type protein. The authors do hint at this in line 435, but the wording of the sentence cited above could be somewhat misleading. Is it possible that the observed difference in JAK binding compared to IRK binding in the crystal structures is partly a function of crystallizing with short peptides rather than the full-length protein? A discussion of this possibility should be included.

We agree with the reviewer here. We have included an explicit statement on the limitations of our approach (the same approach used by all previous structural and biochemical studies of this nature) in the discussion.

“Our results highlighted differences in interactions of PTP1B with the IRK and JAK activation loops. However, by necessity our structural and biochemical analyses (as well as previous studies by other groups) used synthetic activation loop peptides as substrates, rather than intact JAK and IRK proteins. The limitations of using a peptide, rather than the entire protein, must be noted and it may be that regions outside the activation loop in the intact protein will drive substrate specificity. Therefore, it is not conclusive that these results can be extrapolated to represent the situation *in vivo*. A detailed analysis of PTP1B with full-length JAK and IRK proteins is required to gain a full understanding of PTP1B with its substrates and our future efforts lay in this direction.”

Minor comments:

Ln 104 – Italicize *in vitro* corrected “To confirm that the activation loop of the JAKs are indeed a target of PTP1B, dephosphorylation assays with activated recombinant JAK kinase domains were performed *in vitro*.”

Ln 147 – “hydrolyses” should be “hydrolyzes” corrected “receiving a proton from the water molecule that hydrolyzes the phosphoenzyme intermediate”

Ln 223 – “...with Phe182 is partially...” delete “is” corrected “tyrosyl-peptide is still there and because the WPD-loop is closed with Phe182 partially blocking it from solvent.”

Ln 319 – “then” should be “the” corrected “The second pTyr is dephosphorylated at a ~3-

fold higher rate than the first. In contrast, when the IRK peptide”

Ln 322 – “effected” should be “affected” corrected As our structures indicated the involvement of Arg47 in binding the first pTyr outside the catalytic pocket we tested whether mutation of that arginine affected pTyr preference.”

Ln 475 – “DNA encoding human the WT human...” delete the first “human” Deleted “DNA encoding the WT human PTP1B phosphatase domain”

Ln 477 – “clone” should be “cloned” Corrected His tag was cloned into a pPROEX vector and transformed into BL21 (DE3) *E. coli* cells and expression was induced by addition of 1 mM IPTG at 18°C overnight.”

Ln 480 – “mLn20 mM Tris” something needs to be corrected n was deleted, should have been a space “Cells from 1L culture were resuspended in 40 mL 20 mM Tris (pH 8.0), 5 mM imidazole, 250 mM NaCl, 1 mM PMSF, 2 mM TCEP, and 20 mg lysozyme and lysed by sonication.”

REVIEWERS' COMMENTS:

Reviewer #1 (Remarks to the Author):

I have no other questions.

Reviewer #2 (Remarks to the Author):

All my concerns have been resolved completely.